# Graph Robustness Benchmark: Rethinking and Benchmarking Adversarial Robustness of Graph Neural Networks

**Qinkai Zheng**[†]**, Xu Zou**[†]**, Yuxiao Dong**[‡]**, Yukuo Cen**[†]**, Da Yin**[†]**, Jie Tang**[†]

[†] Department of Computer Science and Technology, Tsinghua University
[‡] Facebook AI
{qinkai, jietang}@tsinghua.edu.cn
{zoux18, cyk20, yd18}@mails.tsinghua.edu.cn
ericdongyx@gmail.com

## Abstract

Recent studies have shown that Graph Neural Networks (GNNs) are vulnerable to adversarial attacks. Previous attacks and defenses on GNNs face common problems like scalability or generality, which hinder the progress of this domain. By rethinking limitations in previous works, we propose Graph Robustness Benchmark (GRB), the first benchmark that aims to provide scalable, general, unified, and reproducible evaluation on adversarial robustness of GNNs. GRB includes (1) scalable datasets processed by a novel splitting scheme; (2) diverse set of baseline methods covering GNNs, attacks, and defenses; (3) unified evaluation pipeline that permits a fair comparison; (4) modular coding framework that facilitates implementation of various methods and ensures reproducibility; (5) leaderboards that track the progress of the field. Besides, we propose two strong baseline defenses that significantly outperform previous ones. With extensive experiments, we can fairly compare all methods and investigate their pros and cons. GRB is open-source and maintains all datasets, codes, leaderboards at `https://cogdl.ai/grb/home`, which will be continuously updated to promote future research in this field.

## 1 Introduction

Graph Neural Networks (GNNs), starting from Graph Convolutional Network (GCN) [1], to a large group of more recent variants [2, 3, 4], have shown promising performance in graph machine learning (ML) tasks in various domains including recommender systems [5], academic network analysis [2], knowledge graphs [6] and molecular graph learning [7]. However, neural networks are known to be vulnerable to adversarial examples [8], and recent works [9, 10, 11, 12] show that GNNs are no exception. Typically, GNNs take an attributed graph as the input, and use the message passing scheme [13] to extract relational information. Attackers may modify the graph structure by adding/removing edges [14, 15], or modify the features of nodes with tiny perturbations [10, 11, 12], or even inject malicious nodes [16, 17] to conduct adversarial attacks on GNNs. These attacks can significantly destroy the performance of GNNs with only small changes to the graph [10].

Threatened by adversarial attacks, researchers have begun to take robustness into consideration while designing new GNNs. New architectures like RobustGCN [18], GRAND [19], ProGNN [20] are designated to improve robustness against adversarial attacks. Other methods, like GNN-SVD [21] or GNNGuard [22], try to alleviate the impact of attacks through preprocessing based on the intrinsic properties of the graph. Despite previous works, there are still several common limitations from both the attacker side and the defender side:

Submitted to the 35th Conference on Neural Information Processing Systems (NeurIPS 2021) Track on Datasets and Benchmarks. Do not distribute.

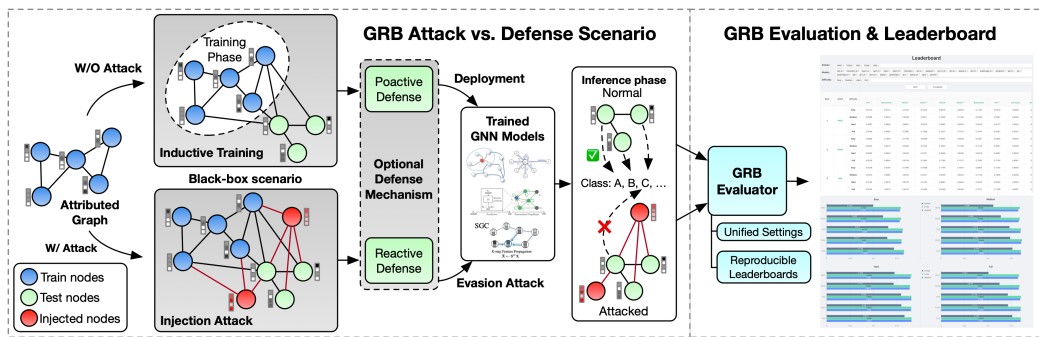

Figure 1: Process of Graph Robustness Benchmark (GRB). It introduces a refined threat model considering: *Black-box*: the attacker only has access to the attributed graph but not the target model; *Evasion*: the attack happens during model inference; *Inductive*: the target model is trained in an inductive setting (test nodes are unseen during training); *Injection*: the attacker is allowed to inject new nodes without modifying existing nodes. All attacks and defenses are evaluated under unified settings and the results are shown on GRB leaderboards.

1. **Ill-defined threat model.** Previous works attempt to imitate the threat model from other domains, like adversarial attacks in image classification, which is not actually suitable for graph structured data. For example, the definition of "unnoticeability" needs to be refined.

2. **Lack of fair comparison among various methods.** Previous works use different settings in their experiments (datasets, data splitting, attack constraints, etc.), each introduces its own bias to the evaluation results, making it difficult to fairly compare the effectiveness of different methods.

3. **Lack of scalability and generality.** Most of previous works only consider small-scale graphs (tens of thousands of nodes), which is far from the scale of real-world applications. Moreover, the assumption of attacking/defending only a single type of defense/attack lacks generality.

Because of these limitations, so far, there is no benchmark on evaluating the *adversarial robustness* of GNNs, i.e. the robustness in the presence of adversarial attacks. Nevertheless, it is an important but challenging task, which requires avoiding pitfalls in previous works and proposing a better solution. Given that there already exist several benchmarks focusing on evaluating the performance of GNNs, like Open Graph Benchmark (OGB) [23] and GNN benchmarking framework [24], it is essential to construct a well-defined, general and scalable graph robustness benchmark.

In this paper, we first revisit the adversarial robustness of GNNs in a principled way. Then, we propose a new benchmark, Graph Robustness Benchmark (GRB). The main goal of GRB is to provide a fair evaluation for adversarial attacks & defenses on GNNs under unified settings. As illustrated in Figure 1, GRB is designed to include the following features:

1. **Refined threat model.** GRB provides a refined threat model and gives precise definitions of attacker and defender's capabilities. The framework clarifies the information and possible actions for both sides, resulting in a new challenging and realistic evaluation scenario.

2. **Elaborated datasets.**[1] GRB consists of five datasets with different scales. The datasets are under an innovative splitting scheme which helps to better evaluate the adversarial robustness of GNNs under different levels of difficulties.

3. **Unified evaluation pipeline.** GRB provides a unified evaluation pipeline that calibrates the experiment settings, which helps to make fair comparisons for both attacks and defenses.

4. **Reproducible leaderboards.**[2] GRB offers leaderboards for each dataset as well as codes, trained models, attack results, and scripts that help to easily reproduce all results. The leaderboards are continuously updated and maintained to ensure reproducibility and to track the progress of adversarial robustness researches on GNNs.

---

[1]https://cogdl.ai/grb/datasets
[2]https://cogdl.ai/grb/leaderboard

5. **Extendable coding framework.**[3] GRB has an extendable framework supporting all above features. GRB also contains implementations of existing methods and is based on a modular design that facilitates researchers to add new GNN models, attacks, or defenses conveniently.

6. **Abundant baseline methods.** GRB currently has diverse set of baseline methods covering GNNs, attacks, and defenses. We also propose two general defense mechanisms that can significantly improve the robustness of GNNs and regard them as strong baselines. GRB will be continuously be elaborated with more methods as the research advances.

Overall, GRB serves as a *scalable*, *general*, *unified*, and *reproducible* benchmark on adversarial robustness of GNNs. We believe that it can help researchers to investigate pros and cons of previous works, and provide insights for future research.

## 2 Rethinking Adversarial Robustness in Graph ML

### 2.1 General Definition

In graph ML, adversarial robustness refers to the performance of graph ML models (e.g. GNNs) under potential adversarial attacks. Take node classification as an example, for an attributed graph $\mathcal{G} = (\mathcal{A}, \mathcal{F})$ where $\mathcal{A} \in \mathbb{R}^{N \times N}$ represents the adjacency matrix of $N$ nodes and $\mathcal{F} \in \mathbb{R}^{N \times D}$ the set of node features with $D$ dimensions for every node. Define a graph model $\mathcal{M} : \mathcal{G} \rightarrow Z$ where $Z \in [0, 1]^{N \times L}$, that maps a graph $\mathcal{G}$ to probability vectors of all $N$ nodes across $L$ classes. The adversarial attack can be formulated as:

$$\max_{\mathcal{G}'} |\mathcal{M}(\mathcal{G}') \neq \mathcal{M}(\mathcal{G})| \tag{1}$$

where $\mathcal{G}' = (\mathcal{A}', \mathcal{F}')$ is the graph modified by attackers. The attacker try to maximize the number of wrong predictions. Usually, there is an assumption that the attack should be *unnoticeable* [9]:

$$d_{\mathcal{A}}(\mathcal{A}', \mathcal{A}) \leq \Delta_{\mathcal{A}} \quad and \quad d_{\mathcal{F}}(\mathcal{F}', \mathcal{F}) \leq \Delta_{\mathcal{F}} \tag{2}$$

where $d_{\mathcal{A}}$ and $d_{\mathcal{F}}$ are the functions that measure the changes between $\mathcal{A}'$ and $\mathcal{A}$, $\mathcal{F}'$ and $\mathcal{F}$. These changes are limited by the constraints $\Delta_{\mathcal{A}}$ and $\Delta_{\mathcal{F}}$. Although this kind of definition is frequently used in previous works, we discuss in the following parts that there are actually some pitfalls.

### 2.2 Revisiting Adversarial Attacks on GNNs

In the domain of security, it is essential to define a threat model, which determines the capability of the attackers. As shown in Table 1, we categorize adversarial attacks on GNNs into several types. Some of terms (*Black-box / White-box*) are inherited from adversarial attacks in image classification [27], others (*Poisoning / Evasion* [9], *Modification / Injection* [17], *Transductive / Inductive* [2]) are specific for graph structured data and GNNs. Here, we give precise definitions of each term:

Table 1: Categorization of adversarial attacks on GNNs. GRB supports the implementation of all kinds of attacks. For GRB leaderboard, we mainly consider the following case: *Black-box*, *Evasion*, *Inductive*, *Injection*.

| Attack | Knowledge | | Objective | | Approach | | Training | |
|---|---|---|---|---|---|---|---|---|
| | *Black.* | *White.* | *Poi.* | *Eva.* | *Mod.* | *Inj.* | *Trans.* | *Ind.* |
| **DICE** [14] | ✓ | | ✓ | | ✓ | | ✓ | |
| **FGSM** [9] | ✓ | ✓ | ✓ | | ✓ | | ✓ | |
| **RND** [9] | ✓ | | ✓ | | ✓ | | ✓ | |
| **Nettack** [9] | ✓ | ✓ | ✓ | | ✓ | | ✓ | |
| **RL-S2V** [25] | ✓ | ✓ | ✓ | | ✓ | | ✓ | |
| **Metattack** [10] | ✓ | | ✓ | | ✓ | | ✓ | |
| **PGD-Topo** [15] | ✓ | ✓ | ✓ | | ✓ | | ✓ | |
| **AFGSM** [16] | ✓ | | ✓ | | | ✓ | ✓ | |
| **SPEIT** [26] | ✓ | | | ✓ | | ✓ | ✓ | |
| **TDGIA** [17] | ✓ | | | ✓ | | ✓ | | ✓ |
| **GRB Support** | ✓ | ✓ | ✓ | ✓ | ✓ | ✓ | ✓ | ✓ |
| **GRB Leaderboard** | ✓ | | | ✓ | | ✓ | | ✓ |

**Attacker's knowledge.** *Black-box*: The attackers do **NOT** have access to the targeted model (including its architecture, parameters, defense mechanism, etc.). However, they have access to the graph data (structure, features, labels of training data, etc.). Besides, they are allowed to query the GNNs and get the outputs. *White-box*: The attackers have access to **ALL** information as the defender has. However, if the targeted model has random process, the runtime randomness should not be available for the attackers.

**Attacker's objective.** *Poisoning*: The attackers generate corrupted graph data and assume that the targeted model is (re)trained on these data to get a worse model. *Evasion*: The attackers generate corrupted graph data to affect the runtime performance of a trained model.

**Attacker's approach.** *Modification*: The attackers modify the original graph (the same one used by the defenders for training) by adding/removing edges or perturbing the value of features. *Injection*:

---
[3]https://github.com/THUDM/grb

The attackers do not modify the original graph but inject new malicious nodes to influence the nodes in the original graph.

**GNNs' training approach.** *Transductive*: The targeted model is trained with the entire graph containing all nodes (including training, validation, test nodes). *Inductive*: The targeted model is trained with the graph containing only the training nodes.

As show in Table 1, previous works cover various combination of these categories. However, there are some common limitations: (1) **Lack of scalability**: most attacks only evaluate in very small graphs and are not scalable to large ones. (2) **Lack of generality**: most attacks only evaluate on basic GCNs, without showing effectiveness to other kinds of GNNs or in the presence of defenses. (3) **Ill-defined threat model**: the threat model in some works is actually ill-defined, especially for the *poisoning* attack under *transductive* training setting, which will be explained in the Section 2.4.

## 2.3 Revisiting Defenses for GNNs

The defenses for GNNs can mainly be categorized into two types: *Preprocess-based* and *Model-based*. In the case of an attributed graph, the defender can preprocess the adjacency matrix (e.g. GNN-SVD [21], GNN-Jaccard [28]) or the features of nodes (e.g. feature transformation [26]). Robustness can also be achieved through *model enhancement*, either by robust training scheme (e.g. adversarial training [29, 30]) or new model architectures (e.g. RobustGCN [18], GNNGuard [22]). Despite many attempts of defenses, they have some common limitations: (1) **Lack of scalability**: defenses are not scalable to large graphs due to time/memory complexity. (2) **Lack of generality**: defenses are proposed to defend only certain types of GNNs with ad-hoc designs, or are only effective against certain types of attacks. (3) **Fragmented evaluation**: there are many biases (choice of datasets, random splitting, various threat models, choice of attacks, different constraints, etc.) introduced in the evaluation process, making it hard to compare the effectiveness of different defenses.

## 2.4 Rethinking the Notion of *Unnoticeability*

Many of the previous adversarial attacks [9, 25, 10] consider the *poisoning* attack and develop the notion of *unnoticeability*, similar to Eq. 2. The initial idea is to imitate the same notion in image classification task: the differences of adversarial examples, compared with clean examples, should be tiny and unnoticeable, so that humans can still easily recognize the objects in images. That's why $l_p$-norm is a widely-used constraint, as it corresponds to the visual sense of humans.

In the *poisoning* setting of graph modification attacks, the attackers assume that the graph is perturbed with corrupted nodes and edges, in a way that the perturbed graph is close to the original one. However, this assumption is controversial: If defenders have the original graph, they can simply train the model on that one; If defenders do not have the original graph (the general case for data poisoning where defenders can not tell whether the data are benign or not), then it does not make sense to keep *unnoticeability*. In this case, we only have $\mathcal{G}' = (\mathcal{A}', \mathcal{F}')$ but not $\mathcal{G} = (\mathcal{A}, \mathcal{F})$ in Eq. 2, making it almost impossible to compare them. Previous works propose to compare the graph properties, like degree distribution [9], feature statistics [28] or topological properties [15]. However, all these comparisons need to be done in presence of the original graph. This is different from the case of images, where *unnoticeability* can be easily judged by humans even without ground-truth images.

The attackers may perturb the graph structure or attributes within the scope of *unnoticeability* defined by themselves, while defenders have to depend on their own observations to discover. For example, Nettack [9] proposes to keep the degree distribution of modified graph similar to the original one. However, even if defenders notice that the degree distribution is different, it is still hard to identify specific malicious nodes or edges from the entire graph. On the contrary, defenses like GNNGuard [22] can use the dissimilarity between features to alleviate effects of perturbations. We argue that it is inadequate to simply adopt the notion from image classification, and to make two graphs "similar" in whatever way. Indeed, there is not an absolute definition, but it is recommended that: *"Unnoticeability" shall be considered from the defenders' view instead of the attackers'*.

## 2.5 Unifying Evaluation of Adversarial Robustness for both Attack and Defense

As mentioned in the above sections, there are some common limitations in both attacks and defenses, making it hard to evaluate the adversarial robustness in graph ML. To tackle these problems, we propose a unified evaluation scenario in GRB for fair comparisons between attacks and defenses. As shown in Figure 1, to make it realistic, the scenario is *Black-box*, *Inductive*, *Evasion*, *Injection* (as defined in Section 2.2). Take the case of a citation-graph classification system for example. The

platform collects labeled data from previous papers and trains a GNN model. When a batch of new papers are submitted, it updates the graph and uses the trained model to predict labels for them.

(1) *Evasion*: We assume that the GNNs are already trained in trusted data (e.g. authenticated users), which are untouched by the attackers but might have natural noises. Thus, attacks will only happen during the inference phase. (2) *Inductive*: We assume that the GNNs are used to classify unseen data (e.g. new users), i.e. validation or test data are unseen during training, which requires GNNs to generalize to out of distribution data. (3) *Injection*: We assume that the attackers can only inject new nodes but not modify the target nodes directly. Since it is usually hard to hack into users' accounts and modify their profiles. However, it is easier to create fake accounts and connect them to existing users. (4) *Black-box*: Both the attacker and the defender have no knowledge about the applied methods each other uses. We further clarify attacker and defender's capability in the following:

1. **For attackers**: they have knowledge about the entire graph (including all nodes, edges and labels, **excluding** labels of the test nodes to attack), but do **NOT** have knowledge about the target model or the defense mechanism; they are allowed to inject a limited number of new nodes with limited edges, but are **NOT** allowed to modify the original graph; they are allowed to generate features of injected nodes as long as they remain *unnoticeable* by defenders (e.g. nodes with features that exceed the range can be easily detected); they are allowed to get the classification results from the target model through limited number of queries.

2. **For defenders**: they have knowledge about the entire graph **excluding** the test nodes to be attacked (thus only the training and validation graph); they are allowed to use any method to increase adversarial robustness, but **NOT** having prior knowledge about what kind of attack is used or about which nodes in the test graph are injected nodes.

Besides, it is reasonable that both sides can make assumptions even in *Black-box* scenario. For example, the attackers can assume that the GNN-based system uses GCNs, since it is one of the most popular GNNs. Also, it is not reasonable to assume that the defense mechanism can be completely held secretly, known as the Kerckhoffs' principle [31]. If a defense wants to be general and universal, it should guarantee part of robustness even when attackers have some knowledge about it.

Following the above assumptions, we are able to provide a unified evaluation scenario, in which a fair comparison of attacks and defenses could be done in a principled way. Moreover, *unnoticeability* becomes meaningful in this case because defenders can compare test data with train data, thus attackers need to pay attention to it. We believe that the scenario covers limitations in previous works and helps promote future research in this field. It is worth mentioning that this is not the only scenario, more well-defined scenarios may be introduced according to the progress of the field in the future.

## 3 GRB: Graph Robustness Benchmark

### 3.1 Overview of GRB

GRB is proposed as a benchmark for evaluating the adversarial robustness of GNNs. It enables fair and convenient evaluations for various attacks and defenses, especially in the above-defined scenario. To this end, GRB includes scalable datasets, unified evaluator, and up-to-date leaderboards to track the most recent progress of this domain. Furthermore, GRB has a modular coding framework based on popular deep learning libraries (Figure 2), which is more than a benchmark. This design facilitates implementations of GNN models, attacks, and defenses, which help to ensure reproducibility and extendability for future works.

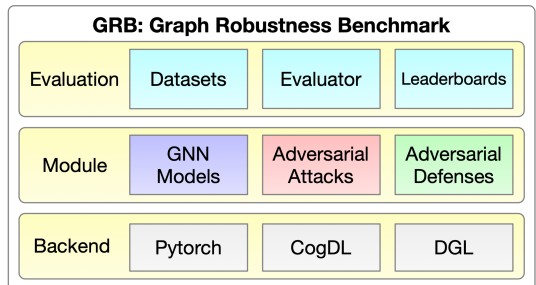

Figure 2: GRB Implementation Framework.

Altogether, GRB serves as a *scalable*, *general*, *unified*, *reproducible*, and *extendable* benchmark on evaluating adversarial robustness of GNNs. In the following subsections, we introduce the implementation of GRB and design details as well as the motivation behind them.

## 3.2 GRB Framework

GRB is mainly built on PyTorch [32], and also supports popular graph learning libraries like CogDL [33] and DGL [34]. It provides a modular coding framework, which allows users to conveniently use the implemented methods, and to add new ones. It contains several modules that support the process introduced in Figure 1: (1) *Dataset*: loads GRB datasets and applies necessary preprocessing including splitting scheme and features normalization; it also allows users to customize their own datasets and make them compatible with GRB evaluation framework. (2) *Model*: implements GNN models, which supports models built on pure Pytorch, CogDL or DGL by automatically transforming the inputs to the required formats. (3) *Attack*: builds adversarial attacks on GNNs, the process of attack is abstracted to different components. For example, graph injection attacks are decomposed to node injection and feature generation. (4) *Defense*: engages defense mechanism to GNN models, including *preprocess-based* and *model-based* defenses. (5) *Evaluator*: evaluates one/multiple methods under unified evaluation settings, i.e. same datasets, constraints and evaluation metrics. (6) *Pipeline*: unifies the entire process of evaluation: load datasets, train/load models, apply attacks/defenses, and finally get the robustness evaluation results; it also helps to easily reproduce the exact results on GRB leaderboards. Apart from these modules, there are also some others like *Trainer* for model training, *Visualise* for visualizing the attack process.

This implementation framework allows GRB to have the following features: (1) *Easy-to-use*: GNN models or attacks can be easily built by only a few lines of codes (Figure 3). (2) *Fair-to-compare*: all methods can be fairly compared under unified settings. (3) *Up-to-date*: GRB maintains leaderboards for each dataset and continuously track the progress of this domain. (4) *Guarantee-to-reproduce*: unlike other benchmarks that just display the results, GRB attaches great importance to reproducibility. For reproducing results on leaderboards, all necessary components are available, including model weights, attack parameters, generated adversarial results, etc. Besides, GRB provides scripts that allow users to reproduce results by a single command line. For all future submissions to GRB, we insist that they should all respect the reproducibility rules detailed in Appendix A.4. All codes are available in `https://github.com/THUDM/grb.` where the implementation details and examples can be found. GRB also provides full documentation for each module and function.

```python
import torch # pytorch backend
from grb.dataset import Dataset
from grb.model.torch import GCN
from grb.utils.trainer import Trainer

# Load data
dataset = Dataset(name='grb-cora', mode='easy',
                  feat_norm='arctan')
# Build model
model = GCN(in_features=dataset.num_features,
            out_features=dataset.num_classes,
            hidden_features=[64, 64])
# Training
adam = torch.optim.Adam(model.parameters(), lr=0.01)
trainer = Trainer(dataset=dataset, optimizer=adam,
                  loss=torch.nn.functional.nll_loss)
trainer.train(model=model, n_epoch=200, dropout=0.5,
              train_mode='inductive')
```

```python
from grb.attack.tdgia import TDGIA

# Attack configuration
tdgia = TDGIA(lr=0.01,
              n_epoch=10,
              n_inject_max=20,
              n_edge_max=20,
              feat_lim_min=-0.9,
              feat_lim_max=0.9,
              sequential_step=0.2)
# Apply attack
rst = tdgia.attack(model=model,
                   adj=dataset.adj,
                   features=dataset.features,
                   target_mask=dataset.test_mask)
# Get modified adj and features
adj_attack, features_attack = rst
```

Figure 3: Code example of GRB. **Left**: Train GCNs on *grb-cora* dataset. **Right**: Apply TDGIA attack on the trained model. GRB facilitates the use of the GNN models, attacks and defenses.

## 3.3 GRB Baseline Methods

GRB currently has a number of implemented methods including GNN models, attacks, and defenses. **GNN models**: GRB includes 7 popular GNN models, GCN [1], GAT [3], GIN [4], APPNP [35], TAGCN [15], GraphSAGE [2], SGCN [36]. Note that these models are not originally designed to increase the robustness. **Attacks**: GRB adapt 5 baseline attacks to the proposed scenario: RND [9], FGSM [8], PGD [29], SPEIT [26], TDGIA [17]. All these methods are implemented as graph injection attacks and are scalable to large-scale graphs. **Defenses**: GRB adopts RobustGCN (R-GCN) [18], GNN-SVD [21] and GNNGuard [22]. We also find that techniques like layer normalization (LN) [37] and adversarial training (AT) [29], if properly used in the proposed scenario, can significantly increase the robustness of various GNN models and outperform current methods. The proposed LN is to apply LN on the input features and after each graph convolutional layer (except the last layer). The idea is to stabilize the dynamics of input and hidden states to alleviate the impact of adversarial perturbations.

The proposed AT is to apply injection attacks during training to make GNNs more robust. In each iteration, we apply FGSM with a few steps to attack the current model and repeat this until the loss converges. These two defenses are general and scalable, and the experiment results show that they outperform previous methods significantly. Thus, we include them in GRB as strong baselines for defenses. More details of these methods can be found in Appendix A.3.

### 3.4 Datasets

Table 2: Statistics of five datasets in GRB, which cover from small to large scale graphs.

| Dataset | Type | Scale | #Nodes | #Edges | #Features | #Classes |
|---------|------|-------|--------|--------|-----------|----------|
| *grb-cora* | Academic networks | Small | 2,680 | 5,148 | 302 | 7 |
| *grb-citeseer* | Academic networks | Small | 3,191 | 4,172 | 768 | 6 |
| *grb-flickr* | Social networks | Medium | 89,250 | 449,878 | 500 | 7 |
| *grb-reddit* | Social networks | Large | 232,965 | 11,606,919 | 602 | 41 |
| *grb-aminer* | Academic networks | Large | 659,574 | 2,878,577 | 100 | 18 |

**Scalability.** GRB includes five datasets of different scales, *grb-cora*, *grb-citeseer*, *grb-flickr*, *grb-reddit*, *grb-aminer*. The original datasets are gathered from previous works [38, 39, 17] , and in GRB they are reprocessed. The fundamental statistics of these datasets are shown in Table 2. Besides small-scale datasets which are common in previous works, GRB also includes medium and large-scale datasets for hundreds of thousands of nodes and millions of edges. More details about how the datasets are generated can be found in Appendix A.1.

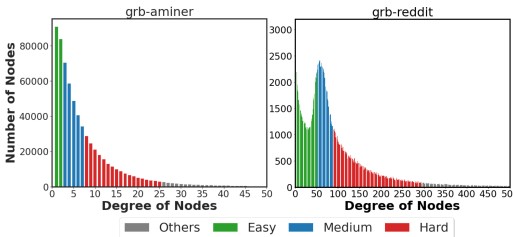
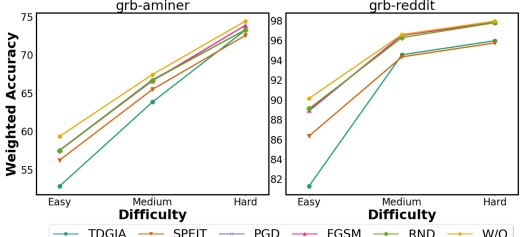

Figure 4: GRB splitting scheme. Difficulties are related to average degree of test nodes.

Figure 5: Effect of dataset difficulties on the performance of adversarial attacks.

**Splitting scheme.** Random splits are not suitable for a fair comparison across methods, especially when it indeed affects the evaluation results of GNNs [40]. GRB introduces a new splitting scheme specially designed for evaluating adversarial robustness. The key idea is based on the assumption that nodes with lower degrees are easier to attack, as demonstrated in [17]. In principle, GNNs aggregate information from neighbor nodes to update a target node. If the target node has few neighbors, it is more likely to be influenced by adversarial perturbations, vice-versa. Thus, we construct test subsets with different average degrees. Firstly, we rank all nodes by their degrees. Secondly, we filter out 5% nodes with the lowest degrees (including isolated nodes that are too easy to attack) and 5% nodes with the highest degree (including nodes connected to hundreds of other nodes that are hardly influenced). Thirdly, we divide the rest nodes into three equal partitions without overlap, and randomly sample 10% nodes (without repetition) from each partition. Finally, we get three test subsets with different degree distributions (Figure 4). According to the average degrees, we define them as Easy/Medium/Hard/Full ('E/M/H/F', 'F' contains all test nodes). For the rest nodes, we divide them into train set (60%) and val set (10%), for training and validation respectively.

**Feature normalization.** Initially, the features in each dataset have various ranges. To make them in the same scale (e.g. range $[-1, 1]$), we apply a *standardization* following by an *arctan* transformation: $\mathcal{F} = \frac{2}{\pi}arctan(\frac{\mathcal{F}-mean(\mathcal{F})}{std(\mathcal{F})})$. Finally, the statistics of datasets after splitting scheme and feature normalization can be found in Appendix A.1.

## 4 Experiments

### 4.1 Experimental Settings

**Methods.** For baseline models, we include 7 popular GNN models, GCN [1], GAT [3], GIN [4], APPNP [35], TAGCN [15], GraphSAGE [2], SGCN [36]. For adversarial attacks, we adapt five

baselines to the proposed scenarios: RND [9], FGSM [8], PGD [29], SPEIT [26], TDGIA [17]. For robustness-enhancement defenses, we adopt RobustGCN (R-GCN) [18], GNN-SVD [21], GNN-Guard [22]. We also include two general methods, layer normalization (LN) [37] and adversarial training (AT) [29] to the proposed scenarios. All details of these methods and hyper-parameters can be found in Appendix A.3 A.4.

**Evaluation metrics.** For attacks: (1) **Avg.**: Average accuracy for various defense models on the attack scenario. (2) **Avg. 3-Max**: Average accuracy for the 3 most robust models (maximum accuracy). (3) **Weighted**: Weighted accuracy across various attacked models, calculated by:$s_w^{att} = \sum_{i=1}^{n} w_i s_i, w_i = \frac{1/i^2}{\sum_{i=1}^{n}(1/i^2)}, s_i = (S_{descend}^{def})_i$ where $S_{descend}^{def}$ is the set of defense scores in a descending order. The metric attaches more weight to the most robust defenses. For defenses: (1) **Avg.**: Average accuracy across various attacks. (2) **Avg. 3-Min**: Average accuracy across the 3 most effective attacks (minimum accuracy). (3) **Weighted**: Weighted accuracy across various attacks, calculated by:$s_w^{def} = \sum_{i=1}^{n} w_i s_i, w_i = \frac{1/i^2}{\sum_{i=1}^{n}(1/i^2)}, s_i = (S_{ascend}^{att})_i$ where $S_{ascend}^{att}$ is the set of attack scores in an ascending order. The metric attaches more weight to the most effective attacks.

## 4.2 Experimental Results

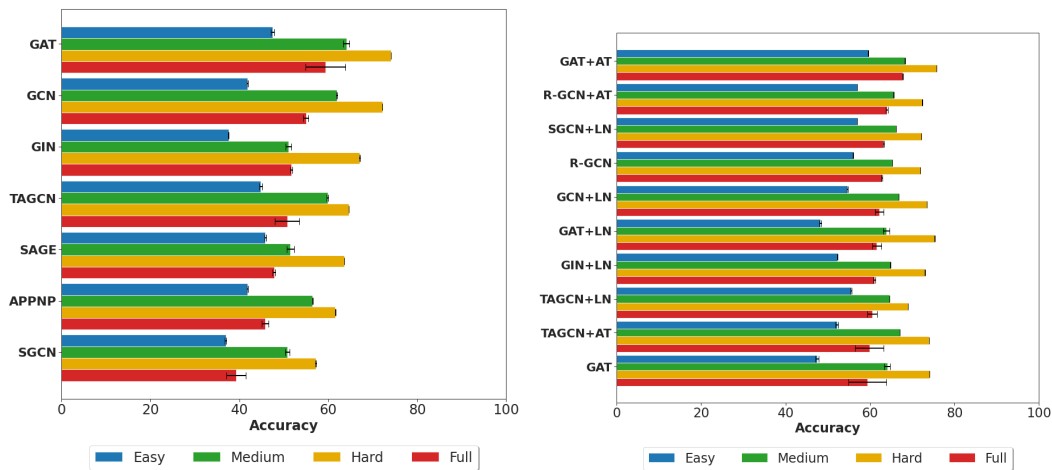

Figure 6: Ranking adversarial robustness of GNNs (W/O Defense) for *grb-aminer* dataset.

Figure 7: Ranking adversarial robustness of GNNs (W/ Defense) for *grb-aminer* dataset.

With GRB, we are able to conduct extensive experiments. We show an example of the GRB leaderboard, and the effect of two proposed defense baselines. More results for all datasets can be found in Appendix A.5 or in our website.

**Example of GRB leaderboard.** Following the GRB process in Figure 1, we evaluate the performance of attacks vs. defense. Table 3 shows a example of leaderboard for *grb-aminer* dataset. The attacks are repeated 10 times to report the error bar. Both attacks and defenses are ranked by the weighted accuracy, where red and blue indicated the best results in each difficulty. We also compare the adversarial robustness of GNNs with or without defense (Figure 7 and 6). Clearly, the defense can help to significantly improve the robustness of GNNs.

**Effect of average degrees of test nodes.** The new splitting scheme is designed to investigate the effect of average degree of test nodes on the attack performance. As shown in Figure 5, attacks tend to better decrease the accuracy for nodes with lower average degree (Easy), which confirms the assumption the adversarial robustness of GNNs is related to the degree of nodes.

**Effect of layer LN and AT.** Figure 8 and 9 shows that the proposed LN and AT can generally increase the robustness of various types of GNNs. The detailed algorithms can be found in Appendix A.3.3. The leaderboards also show that GNNs defended by these two methods have SOTA adversarial robustness compared with previous works.

Table 3: GRB leaderboard (Top 5 Attacks vs. Top 10 Defenses) for *grb-aminer* dataset.

| Attack | | Defenses 1 GAT+AT | 2 R-GCN+AT | 3 SGCN+LN | 4 R-GCN | 5 GCN+LN | 6 GAT+LN | 7 GIN+LN | 8 TAGCN+LN | 9 TAGCN+AT | 10 GAT | Avg. Accuracy | Avg. 3-Max Accuracy | Weighted Accuracy |
|---|---|---|---|---|---|---|---|---|---|---|---|---|---|---|
| 1 TDGIA | E | 59.54$_{\pm.05}$ | 56.83$_{\pm.06}$ | 56.73$_{\pm.06}$ | 56.12$_{\pm.07}$ | 53.51$_{\pm.21}$ | 43.93$_{\pm.41}$ | 51.10$_{\pm.12}$ | 54.63$_{\pm.20}$ | 49.59$_{\pm.50}$ | 42.40$_{\pm.52}$ | 52.44$_{\pm.17}$ | 57.70$_{\pm1.31}$ | 58.08$_{\pm.04}$ |
| | M | 68.39$_{\pm.02}$ | 65.61$_{\pm.02}$ | 66.11$_{\pm.02}$ | 65.23$_{\pm.03}$ | 66.78$_{\pm.05}$ | 61.84$_{\pm1.20}$ | 64.49$_{\pm.10}$ | 64.62$_{\pm.02}$ | 67.27$_{\pm.04}$ | 62.47$_{\pm1.01}$ | 65.28$_{\pm.23}$ | 67.48$_{\pm.68}$ | 67.69$_{\pm.02}$ |
| | H | 75.83$_{\pm.02}$ | 72.35$_{\pm.02}$ | 72.10$_{\pm.00}$ | 71.94$_{\pm.02}$ | 73.39$_{\pm.02}$ | 75.22$_{\pm.04}$ | 72.92$_{\pm.02}$ | 68.94$_{\pm.03}$ | 73.98$_{\pm.01}$ | 75.03$_{\pm.03}$ | 73.17$_{\pm.01}$ | 75.36$_{\pm.34}$ | 75.33$_{\pm.01}$ |
| | F | 67.69$_{\pm.03}$ | 63.62$_{\pm.32}$ | 62.20$_{\pm.15}$ | 61.99$_{\pm.22}$ | 60.38$_{\pm1.46}$ | 59.69$_{\pm1.57}$ | 59.59$_{\pm.42}$ | 59.06$_{\pm1.75}$ | 57.24$_{\pm5.04}$ | 56.63$_{\pm6.75}$ | 60.81$_{\pm1.71}$ | 64.52$_{\pm2.32}$ | 65.74$_{\pm.21}$ |
| 2 SPEIT | E | 59.54$_{\pm.07}$ | 56.80$_{\pm.05}$ | 56.94$_{\pm.10}$ | 55.64$_{\pm.10}$ | 56.15$_{\pm.06}$ | 56.13$_{\pm.07}$ | 54.24$_{\pm.09}$ | 56.61$_{\pm.06}$ | 56.59$_{\pm.08}$ | 57.36$_{\pm.09}$ | 56.60$_{\pm.04}$ | 57.95$_{\pm1.14}$ | 58.62$_{\pm.05}$ |
| | M | 68.37$_{\pm.04}$ | 65.46$_{\pm.03}$ | 66.20$_{\pm.02}$ | 65.25$_{\pm.05}$ | 66.75$_{\pm.03}$ | 67.49$_{\pm.06}$ | 65.05$_{\pm.06}$ | 64.47$_{\pm.04}$ | 66.95$_{\pm.05}$ | 66.81$_{\pm.04}$ | 66.28$_{\pm.02}$ | 67.60$_{\pm.59}$ | 67.86$_{\pm.03}$ |
| | H | 75.94$_{\pm.04}$ | 72.27$_{\pm.03}$ | 72.36$_{\pm.03}$ | 71.86$_{\pm.03}$ | 73.41$_{\pm.01}$ | 75.34$_{\pm.03}$ | 72.87$_{\pm.03}$ | 68.88$_{\pm.05}$ | 73.98$_{\pm.02}$ | 73.83$_{\pm.04}$ | 73.07$_{\pm.01}$ | 75.08$_{\pm.82}$ | 75.33$_{\pm.02}$ |
| | F | 68.04$_{\pm.03}$ | 64.05$_{\pm.04}$ | 64.84$_{\pm.04}$ | 64.06$_{\pm.04}$ | 65.51$_{\pm.02}$ | 64.02$_{\pm.04}$ | 63.11$_{\pm.02}$ | 62.59$_{\pm.04}$ | 63.77$_{\pm.06}$ | 63.58$_{\pm.06}$ | 64.36$_{\pm.02}$ | 66.13$_{\pm1.38}$ | 66.89$_{\pm.02}$ |
| 3 RND | E | 59.56$_{\pm.06}$ | 57.53$_{\pm.06}$ | 57.41$_{\pm.06}$ | 56.38$_{\pm.11}$ | 57.76$_{\pm.05}$ | 58.83$_{\pm.10}$ | 54.41$_{\pm.13}$ | 58.07$_{\pm.12}$ | 58.14$_{\pm.04}$ | 57.46$_{\pm.10}$ | 57.55$_{\pm.03}$ | 58.85$_{\pm.57}$ | 59.09$_{\pm.05}$ |
| | M | 68.22$_{\pm.04}$ | 65.86$_{\pm.03}$ | 66.29$_{\pm.03}$ | 65.34$_{\pm.06}$ | 67.03$_{\pm.03}$ | 68.62$_{\pm.05}$ | 65.54$_{\pm.06}$ | 64.98$_{\pm.08}$ | 67.34$_{\pm.04}$ | 67.71$_{\pm.06}$ | 66.69$_{\pm.02}$ | 68.18$_{\pm.38}$ | 68.24$_{\pm.03}$ |
| | H | 75.75$_{\pm.02}$ | 72.66$_{\pm.02}$ | 72.42$_{\pm.03}$ | 72.00$_{\pm.03}$ | 73.52$_{\pm.02}$ | 75.63$_{\pm.03}$ | 73.36$_{\pm.03}$ | 69.30$_{\pm.06}$ | 74.04$_{\pm.02}$ | 75.36$_{\pm.03}$ | 73.40$_{\pm.01}$ | 75.58$_{\pm.17}$ | 75.39$_{\pm.01}$ |
| | F | 67.72$_{\pm.04}$ | 64.98$_{\pm.04}$ | 65.31$_{\pm.04}$ | 64.45$_{\pm.04}$ | 66.17$_{\pm.02}$ | 67.54$_{\pm.04}$ | 64.36$_{\pm.06}$ | 64.33$_{\pm.03}$ | 66.42$_{\pm.03}$ | 66.23$_{\pm.04}$ | 65.75$_{\pm.02}$ | 67.23$_{\pm.58}$ | 67.34$_{\pm.03}$ |
| 4 PGD | E | 59.70$_{\pm.06}$ | 57.71$_{\pm.05}$ | 57.73$_{\pm.09}$ | 57.19$_{\pm.07}$ | 57.60$_{\pm.08}$ | 57.05$_{\pm.17}$ | 54.69$_{\pm.09}$ | 58.18$_{\pm.07}$ | 58.27$_{\pm.09}$ | 58.46$_{\pm.11}$ | 57.66$_{\pm.05}$ | 58.81$_{\pm.64}$ | 59.14$_{\pm.05}$ |
| | M | 68.40$_{\pm.05}$ | 66.12$_{\pm.02}$ | 66.39$_{\pm.04}$ | 65.67$_{\pm.04}$ | 67.04$_{\pm.03}$ | 68.24$_{\pm.04}$ | 65.64$_{\pm.08}$ | 65.17$_{\pm.05}$ | 67.32$_{\pm.03}$ | 67.85$_{\pm.05}$ | 66.78$_{\pm.04}$ | 68.16$_{\pm.23}$ | 68.12$_{\pm.03}$ |
| | H | 75.83$_{\pm.04}$ | 72.91$_{\pm.02}$ | 72.47$_{\pm.05}$ | 72.18$_{\pm.05}$ | 73.52$_{\pm.02}$ | 75.55$_{\pm.05}$ | 73.58$_{\pm.04}$ | 69.64$_{\pm.05}$ | 73.89$_{\pm.02}$ | 74.34$_{\pm.04}$ | 73.39$_{\pm.01}$ | 75.24$_{\pm.65}$ | 75.36$_{\pm.02}$ |
| | F | 68.01$_{\pm.02}$ | 65.41$_{\pm.02}$ | 65.54$_{\pm.03}$ | 65.05$_{\pm.03}$ | 66.22$_{\pm.02}$ | 66.49$_{\pm.04}$ | 64.63$_{\pm.04}$ | 64.82$_{\pm.04}$ | 66.14$_{\pm.04}$ | 65.86$_{\pm.01}$ | 66.94$_{\pm.76}$ | 67.37$_{\pm.02}$ | |
| 5 FGSM | E | 59.71$_{\pm.04}$ | 57.69$_{\pm.08}$ | 57.62$_{\pm.06}$ | 57.16$_{\pm.08}$ | 57.60$_{\pm.06}$ | 56.97$_{\pm.09}$ | 54.67$_{\pm.08}$ | 58.20$_{\pm.10}$ | 58.23$_{\pm.06}$ | 58.46$_{\pm.07}$ | 57.63$_{\pm.05}$ | 58.81$_{\pm.65}$ | 59.15$_{\pm.04}$ |
| | M | 68.37$_{\pm.02}$ | 66.10$_{\pm.03}$ | 66.38$_{\pm.04}$ | 65.70$_{\pm.05}$ | 67.03$_{\pm.04}$ | 68.27$_{\pm.04}$ | 65.61$_{\pm.08}$ | 65.16$_{\pm.05}$ | 67.30$_{\pm.02}$ | 67.84$_{\pm.07}$ | 66.78$_{\pm.04}$ | 68.16$_{\pm.23}$ | 68.11$_{\pm.02}$ |
| | H | 75.82$_{\pm.04}$ | 72.92$_{\pm.04}$ | 72.48$_{\pm.05}$ | 72.18$_{\pm.05}$ | 73.52$_{\pm.02}$ | 75.55$_{\pm.05}$ | 73.60$_{\pm.04}$ | 69.64$_{\pm.05}$ | 73.90$_{\pm.01}$ | 74.34$_{\pm.04}$ | 73.39$_{\pm.01}$ | 75.23$_{\pm.65}$ | 75.35$_{\pm.02}$ |
| | F | 68.00$_{\pm.04}$ | 65.41$_{\pm.02}$ | 65.54$_{\pm.03}$ | 65.05$_{\pm.03}$ | 66.22$_{\pm.02}$ | 66.50$_{\pm.06}$ | 64.65$_{\pm.04}$ | 64.82$_{\pm.04}$ | 66.34$_{\pm.03}$ | 66.15$_{\pm.06}$ | 66.95$_{\pm.75}$ | 67.37$_{\pm.01}$ | |
| 6 W/O Attack | E | 59.67$_{\pm.00}$ | 58.08$_{\pm.00}$ | 60.22$_{\pm.00}$ | 58.53$_{\pm.00}$ | 58.14$_{\pm.00}$ | 60.78$_{\pm.00}$ | 56.83$_{\pm.00}$ | 59.47$_{\pm.00}$ | 59.62$_{\pm.00}$ | 59.88$_{\pm.00}$ | 59.12$_{\pm.00}$ | 60.29$_{\pm.37}$ | 60.42$_{\pm.00}$ |
| | M | 68.28$_{\pm.00}$ | 66.14$_{\pm.00}$ | 67.11$_{\pm.00}$ | 66.35$_{\pm.00}$ | 67.00$_{\pm.00}$ | 68.98$_{\pm.00}$ | 66.26$_{\pm.00}$ | 65.41$_{\pm.00}$ | 67.53$_{\pm.00}$ | 68.41$_{\pm.00}$ | 67.15$_{\pm.00}$ | 68.56$_{\pm.30}$ | 68.59$_{\pm.00}$ |
| | H | 75.85$_{\pm.00}$ | 73.05$_{\pm.00}$ | 72.69$_{\pm.00}$ | 72.66$_{\pm.00}$ | 73.46$_{\pm.00}$ | 75.64$_{\pm.00}$ | 73.69$_{\pm.00}$ | 69.84$_{\pm.00}$ | 74.10$_{\pm.00}$ | 75.76$_{\pm.00}$ | 73.67$_{\pm.00}$ | 75.75$_{\pm.09}$ | 75.52$_{\pm.00}$ |
| | F | 67.93$_{\pm.00}$ | 65.76$_{\pm.00}$ | 66.68$_{\pm.00}$ | 65.85$_{\pm.00}$ | 66.20$_{\pm.00}$ | 68.47$_{\pm.00}$ | 66.59$_{\pm.00}$ | 64.91$_{\pm.00}$ | 67.08$_{\pm.00}$ | 68.02$_{\pm.00}$ | 66.65$_{\pm.00}$ | 68.14$_{\pm.24}$ | 68.11$_{\pm.00}$ |
| Avg. Accuracy | E | 59.62$_{\pm.02}$ | 57.44$_{\pm.03}$ | 57.77$_{\pm.04}$ | 56.84$_{\pm.04}$ | 56.79$_{\pm.04}$ | 55.62$_{\pm.06}$ | 54.33$_{\pm.04}$ | 57.53$_{\pm.05}$ | 56.74$_{\pm.09}$ | 55.67$_{\pm.10}$ | - | - | - |
| | M | 68.34$_{\pm.01}$ | 65.88$_{\pm.01}$ | 66.41$_{\pm.01}$ | 65.59$_{\pm.02}$ | 66.94$_{\pm.02}$ | 67.24$_{\pm.19}$ | 65.43$_{\pm.04}$ | 64.97$_{\pm.02}$ | 67.28$_{\pm.01}$ | 66.85$_{\pm.18}$ | - | - | - |
| | H | 75.84$_{\pm.01}$ | 72.69$_{\pm.01}$ | 72.42$_{\pm.01}$ | 72.14$_{\pm.02}$ | 73.47$_{\pm.01}$ | 75.49$_{\pm.01}$ | 73.33$_{\pm.02}$ | 69.38$_{\pm.02}$ | 73.98$_{\pm.00}$ | 74.78$_{\pm.02}$ | - | - | - |
| | F | 67.90$_{\pm.01}$ | 64.87$_{\pm.05}$ | 65.02$_{\pm.03}$ | 64.41$_{\pm.04}$ | 65.12$_{\pm.25}$ | 65.45$_{\pm.26}$ | 63.65$_{\pm.07}$ | 63.42$_{\pm.29}$ | 64.53$_{\pm.84}$ | 64.46$_{\pm1.13}$ | - | - | - |
| Avg. 3-Min Accuracy | E | 59.55$_{\pm.04}$ | 57.05$_{\pm.04}$ | 57.02$_{\pm.03}$ | 56.05$_{\pm.07}$ | 55.77$_{\pm.07}$ | 52.33$_{\pm.12}$ | 53.25$_{\pm.07}$ | 56.43$_{\pm.07}$ | 54.77$_{\pm.16}$ | 52.41$_{\pm.17}$ | - | - | - |
| | M | 68.28$_{\pm.01}$ | 65.64$_{\pm.02}$ | 66.20$_{\pm.01}$ | 65.28$_{\pm.03}$ | 66.84$_{\pm.02}$ | 65.85$_{\pm.40}$ | 65.02$_{\pm.04}$ | 64.69$_{\pm.03}$ | 67.17$_{\pm.02}$ | 65.66$_{\pm.34}$ | - | - | - |
| | H | 75.80$_{\pm.02}$ | 72.42$_{\pm.02}$ | 72.29$_{\pm.01}$ | 71.93$_{\pm.02}$ | 73.42$_{\pm.01}$ | 75.36$_{\pm.02}$ | 73.05$_{\pm.02}$ | 69.04$_{\pm.03}$ | 73.92$_{\pm.01}$ | 74.17$_{\pm.03}$ | - | - | - |
| | F | 67.78$_{\pm.02}$ | 64.22$_{\pm.11}$ | 64.12$_{\pm.06}$ | 63.50$_{\pm.08}$ | 64.02$_{\pm.49}$ | 63.39$_{\pm.53}$ | 62.35$_{\pm.14}$ | 61.99$_{\pm.58}$ | 62.44$_{\pm1.69}$ | 62.11$_{\pm2.26}$ | - | - | - |
| Weighted Accuracy | E | 59.53$_{\pm.04}$ | 56.93$_{\pm.04}$ | 56.94$_{\pm.04}$ | 55.93$_{\pm.08}$ | 54.63$_{\pm.14}$ | 48.21$_{\pm.27}$ | 52.23$_{\pm.08}$ | 55.55$_{\pm.14}$ | 52.18$_{\pm.33}$ | 47.45$_{\pm.35}$ | - | - | - |
| | M | 68.25$_{\pm.02}$ | 65.57$_{\pm.02}$ | 66.17$_{\pm.02}$ | 65.28$_{\pm.02}$ | 66.79$_{\pm.02}$ | 63.85$_{\pm.80}$ | 64.77$_{\pm.07}$ | 64.60$_{\pm.03}$ | 67.06$_{\pm.03}$ | 64.07$_{\pm.68}$ | - | - | - |
| | H | 75.78$_{\pm.02}$ | 72.37$_{\pm.02}$ | 72.20$_{\pm.01}$ | 71.92$_{\pm.03}$ | 73.41$_{\pm.01}$ | 75.30$_{\pm.02}$ | 72.98$_{\pm.02}$ | 68.99$_{\pm.04}$ | 73.91$_{\pm.01}$ | 74.08$_{\pm.03}$ | - | - | - |
| | F | 67.73$_{\pm.02}$ | 63.96$_{\pm.21}$ | 63.19$_{\pm.10}$ | 62.80$_{\pm.15}$ | 62.18$_{\pm.98}$ | 61.58$_{\pm1.05}$ | 61.00$_{\pm.28}$ | 60.54$_{\pm1.18}$ | 59.82$_{\pm3.38}$ | 59.37$_{\pm4.53}$ | - | - | - |

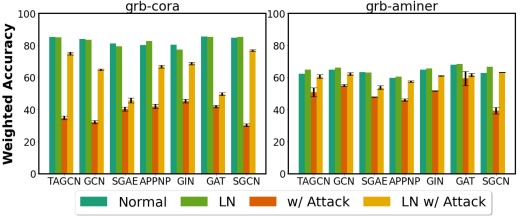

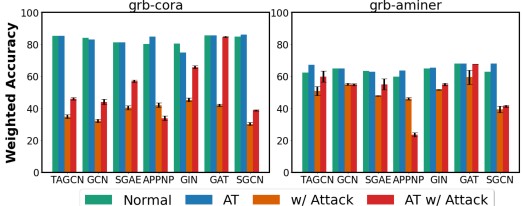

Figure 8: Effect of the proposed layer normalization (LN) on the adversarial robustness of GNNs.

Figure 9: Effect of the proposed adversarial training (AT) on the adversarial robustness of GNNs.

## 5 Conclusion

To improve and facilitate the evaluation of adversarial robustness of GNNs, we rethink limitations in previous works and propose Graph Robustness Benchmark (GRB), a *scalable*, *unified*, *reproducible* and *extendable* benchmark. It has scalable datasets with special design and a unified evaluation pipeline for adversarial robustness. Its coding framework ensures the reproducibility and facilitates the implementation of future methods. We implement various methods and propose two defenses as strong baselines. We believe that GRB is helpful for promoting future research in this field.

## 6 Broader Impact

**Positive impact.** GRB provides a clear and general framework for robustness evaluation of both attacks and defenses. On one hand, it'll help researchers to develop more robust GNNs against adversarial attacks. On the other hand, it'll also help possible attackers to develop better attack methods to turn down defenses. In the case of adversarial attack and defense, the more public information of potential attack and defense methods, the easier the attackers can use public attack methods and the harder he can conduct secret attacks based on private methods. For defenders, the more information about potential attack methods they have, the more generalized robustness defense mechanism can be designed.

**Negative impact.** As this benchmark will offer a lot of public information for both sides, it will also make public attack methods more widely-known and hence GNNs may become more vulnerable. Attackers can also use it to design destructive attacks that may cause damage to GNN-based systems. . GRB also has limitations. It only considers homogeneous graph rather then heterogeneous graph. It focuses on node classification task, while other tasks like link prediction and graph classification are also vulnerable to attacks. Since the domain of adversarial attacks and defenses develops rapidly, we will maintain update GRB continuously to track the progress and we highly welcome the community's contribution to cover these issues in the future.

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
