# OpenReview forum: "Graph Robustness Benchmark: Rethinking and Benchmarking Adversarial Robustness of Graph Neural Networks"
_NeurIPS.cc/2021/Track/Datasets_and_Benchmarks/Round1 — Submitted to NeurIPS 2021 Datasets and Benchmarks Track (Round 1)_

### Official Review · Reviewer_16ob · 2021-06-29
**Not really about benchmarks or datasets**

**Rating:** 4
**Confidence:** 5
**Clarity:** The paper has above-average writing, …

**Strengths:**

The paper is well-written, and raises compelling points in favor of their measure for adversarial robustness. The experiments are thorough, clearly presented, and appear to be easily reproducible as the source code is available and the code style is reasonably clean.

**Weaknesses:**

Unfortunately, this paper is, fundamentally, not about benchmarks and datasets. While the authors do phrase parts of the paper in terms of making a benchmark framework, the content of the paper does not back this up.

For example, there are no comparisons to other works relating to benchmarking in graphs, and appear to be no code interfaces that allow other researchers to employ this new measure of adversarial robustness in standard benchmark frameworks. There is no new data collected, nor any discussion of how this new measure of adversarial robustness could make data collection easier.

The authors discuss a "leaderboard" for the best-performing models, but in actuality, this appears to be little more than a static webpage that displays the experimental results from the paper in a pretty format. (The website itself does not appear to work.)

Finally, there is no compelling argument that this framework will stand the test of time, at least as far as other researchers can use this as a standard benchmark. The authors claim that they will maintain the leaderboard. Perhaps the authors will follow through and update the website, but it honestly seems like that would not be worth their time. The paper is not really about benchmarks or leaderboards; it's about new measures for adversarial robustness in graphs. The leaderboard does not need to be continually updated for the paper to achieve its purpose.

To underscore the above points, I will simply point out that the discussion of actual code frameworks and models does not begin until the end of page 5. This raises the strong presumption that these matters are not the highlight of the paper.

**Additional Feedback:**

I want to stress that this is a good paper. If it had been submitted to the NeurIPS regular track, I would have recommended for acceptance, and if it had been submitted to a workshop or conference devoted to graph representation specifically, I would have given strong support for acceptance.

It is unfortunate to reject this paper on the basis that it was submitted to the wrong venue, but I believe accepting this paper would weaken the the Datasets and Benchmarks track. I encourage the authors to submit their work elsewhere, as I'm sure it will find an interested audience.

**Correctness:**

The datasets and models used in this paper appear to be very standard in the literature on graph learning.

**Documentation:**

The authors use standard datasets and models for their experiments, and these are cited and linked to appropriately.

**Ethics:**

The authors do not introduce any new datasets, and their models and methods are standard, leaving little room for ethical conflicts to arise at all.

**Relation To Prior Work:**

The authors give a reasonably thorough exposition on previous measures and models for attaining adversarial robustness in graph neural networks. They essentially give no discussion of related benchmarking frameworks for graphs.

**Summary And Contributions:**

The paper outlines flaws in existing measures of adversarial robustness in graph neural networks, namely relating to unrealistic assumptions that are typically made about the knowledge of the attacker and the defender. The authors propose a new measure of adversarial robustness which appears to be more suited for graph learning tasks.

In this model, the attackers are allowed to inject new nodes and edges into the network, but the attacker cannot modify existing nodes or node attributes or edges. This is to reflect the realistic attack of creating spam accounts to manipulate the defender's model toward various ends. The defenders have full knowledge of their training and validation set, but not the test set.

The authors then train several standard graph neural network models on a number of standard datasets using their measure of adversarial robustness, and summarize the results.

---

> ### Author Response · Authors · 2021-07-11
> **Thanks for your review. About the nature of GRB and the scope of this track.**
>
>  Please find our responses to your concerns below:
>
> 1. **GRB is a benchmark for adversarial robustness of GNNs.**
>
>    * First, we need to highlight that this track is not only about collecting new datasets, it is also about benchmarks. According to the guidelines of this track (https://nips.cc/Conferences/2021/CallForDatasetsBenchmarks), the datasets and benchmarks are clearly two separated possibilities (of course, they can occur simultaneously). Our work of GRB focuses on the side of the benchmark.
>    * Second, the motivation for designing GRB is that there is currently no standard benchmark for evaluating adversarial robustness. As discussed in Sec 2, previous works have common flaws: lack of scalability, lack of generality, ill-defined threat model, and fragmented evaluation settings. These flaws make it extremely difficult to compare various methods (both attacks and defenses), nor to track the progress of the field.
>    * As a solution, we propose GRB, a benchmark that provides scalable, general, unified, and reproducible evaluation on the adversarial robustness of GNNs. We mentioned datasets in Sec 3.4, but it is not about collecting new data. We can actually adapt any graph data to the proposed scenario, and we processed five datasets of different scales because they can help to better investigate the scalability. We focus more on establishing a standard benchmark. In Sec 2, we proposed a well-defined scenario where the settings for attackers and defenders are unified, which makes it possible to do standard evaluations and comparisons between different methods. Moreover, we further design a modular code framework to ensure reproducibility, which many other benchmarks missed. As for leaderboards, they show the ranking of current or future methods in the literature, which helps us to track the progress of the field, which facilitates future researches.
>
>    So overall, GRB is clearly a benchmark and was designed to be one. We believe GRB suits perfectly the scope of this track and that's why we didn't submit it to the NeurIPS regular track.
>
> 2. **GRB is different from other graph learning benchmarks.** As discussed in the Introduction (line 42-47), there are other benchmarks for graph learning tasks like Open Graph Benchmark [23] or GNN benchmarking framework [24], while they were designed for evaluating the performance of graph learning models. They have nothing to do with the "adversarial robustness" and were not designed to be. Instead, GRB focuses on evaluating the adversarial robustness of GNNs in a principled way. GRB is not a new measure, it consists of a well-defined threat model, a complete evaluation pipeline, a code framework that supports it (for attacks, defenses, and GNNs, while other benchmarks only consider GNNs). The purpose is definitely not to employ GRB to current benchmarks (other benchmarks do not support attacks or defenses and have a totally different evaluation process), however, it is itself a benchmark.
>
> 3. **Role of GRB website.** We recommend you to revisit our website (https://cogdl.ai/grb/home), it seems that you had some problem visiting it. The website is not a simple static webpage that only shows some results as you thought, it consists of the following essential functions:
>
>    * **Documentation**: there are introductions, instructions of how to use it, details of datasets and ways to download them, rules of evaluation to obey, and other detailed documentation.
>    * **Leaderboard**: GRB provides leaderboards that compare attacks vs. defenses, details of how the results were obtained, as well as scripts to easily reproduce all results. Researchers can rely on these codes to test their methods and track the progress of SOTA methods.
>    * **Submission of new methods**: Researchers can submit new methods via google form (in page: https://cogdl.ai/grb/leaderboard/). For new submissions, we highly respect the reproducibility, they should provide the implementations and configurations following the instructions, instead of only submitting results. As the domain develops, there will appear more methods, we will continuously update the website to track the progress. We believe this worth the time and hope it can help the community go further.
>
> We hope these responses can address your concerns.

---

> > ### Comment · Reviewer_16ob · 2021-07-20
> > **Additional Comments**
> >
> > I thank the authors for offering their perspective. I'd perhaps like to clarify mine.
> >
> > I believe the focus of the conference is on enabling other researchers to develop or utilize benchmarks of their own, in a more general sense. This paper solves a problem about adversarial robustness. This conference is, fundamentally, about lowering the difficulty for scientists to make machine learning research more reproducible and experiments of higher quality in general. It's simply a different focus.
> >
> > That said, I believe my initial assessment of "clear reject" was a bit strong. I have upgraded my review to a more neutral "reject."

---

### Official Review · Reviewer_fGSw · 2021-07-04
**Well designed benchmark, concerns about the significance of the chosen task**

**Rating:** 5
**Confidence:** 3
**Correctness:** The evaluation methods seem appropriate.
**Clarity:** The presentation is clear and easy to…

**Strengths:**

- The authors clearly present the different threat models of adversarial robustness in graph ML, and discuss the current limitations about both the constraints imposed on the attacks (the notion of unnoticeable) and the methodology in the evaluation. Based on such analysis, they propose a well-defined scenario, with reasonable assumptions for both attackers and defenders, which allows to compare different techniques in a standardized way. I think such discussion is important as a clear task on which comparing methods might lead to better understanding of existing methods and promote development of new ones.

- GRB provides tools to reproduce the results on the leaderboards and to use attacks and defenses to test new methods. This, together with the new splitting schemes of the datasets, should facilitate the evaluation of new methods and comparison to SOTA.

- The authors propose two new baselines for defenses, applying adversarial training and layer normalization to the existing models, achieving higher robustness the current techniques.

- The interactive leaderboards allow to easily compare attacks and defenses along different metrics.

**Weaknesses:**

- Currently, the benchmark is limited to a single threat model (black-box, evasion, inductive, injection). Although this seems the easiest for a standardized evaluation, I'm not sure how popular or established it is: in fact, according to Table 1 none of the existing attacks focuses on it, and the baselines are adapted (L250). Moreover, the fact that a simple baseline as adversarial training, which is the standard in image classification tasks, hasn't been used in prior works while achieving the best results suggests that this threat model hasn't received much attention.

- The constraints for the attacks should be clearly stated in the main part (I think they're only mentioned in the appendix), especially given the discussion in Section 2. How are they chosen?

- From Figures 6 and 7 and leaderboards it seems that 1) the standard models are already quite robust, 2) the difference in robustness from the most robust model GAT+AT and the standard counterpart GAT is not large (~11% if I read the leaderboard right), 3) the robust accuracy is not particularly lower than the clean one for many models. Then, wouldn't be more appropriate to have stronger attacks, e.g. with the possibility of injecting more nodes or edges, for a more challenging task? It is not clear currently how large the margin for possible improvements is.

**Additional Feedback:**

As mentioned above, my main concerns are about the significance of the chosen threat model.

**Documentation:**

The code to reproduce the experiments is provided.

**Ethics:**

There are no ethical concerns.

**Relation To Prior Work:**

The authors discuss existing works and how different those are from the proposed GRB.

**Summary And Contributions:**

The paper Graph Robustness Benchmark (GRB) to evaluate the performance of adversarial attacks and defenses for GNNs. It first discusses the limitations of the current comparisons, then defines a realistic threat model which is the the object of the benchmark, and finally introduces the structure of the new benchmark and its entries. GRB provides tools to compare new methods in a standardized way. In addition to existing methods, the authors also propose two novel baselines as defenses, which improve the robustness across datasets and models.

---

> ### Author Response · Authors · 2021-07-11
> **Thanks very much for your valuable comments. About the choice of the threat model.**
>
> Please find our responses to your concerns below:
>
> 1. **Choice of threat model**: For the chosen threat model, we need to clarify that it is not a random choice or a simplification. As discussed in Sec 2, we pointed out the problems of ill-defined threat models, and finally choose **the most consistent and realistic one** from all the combinations for the following reasons:
>
>    * **black-box vs. white-box**: black-box is the most common scenario for real-world applications. It also encourages researchers to consider more general attacks that are effective for various GNNs, instead of attacks that only work when knowing details of specific defenses.
>    * **evasion vs. poisoning**: as in Se 2.4, we found most of works follow the setting of poisoning attack in [9], which develops the term of *unnoticeability*, while this setting is controversial. Under the poisoning setting, defenders do not have a clean graph and do not know whether the data are polluted or not. In this case, one can modify the graph largely while not being detected, like simply using large perturbation or label shuffling. Previous works claimed using small perturbations, but the truth is even attackers use much larger ones, the perturbed nodes can hardly identify under the poisoning setting (This is quite different from images, which can be easily judged by visual sense. Even there are topological properties for graph data, it is difficult to tell whether individual nodes are polluted or not). Thus, there is not yet reasonable constraint or realistic scenario for poisoning attack.
>    * **injection vs. modification**: in reality, if attackers want to modify data on individual nodes, they need to hack them to get the authority, which is difficult. Instead, it is much easier to inject fake nodes (e.g., fake accounts or papers) into existing nodes. This makes injection attack a more realistic scenario.
>    * **inductive vs. transductive**: to make the *unnoticeability* meaningful and to calibrate the capability of attackers and defenders, we assume that GNNs are trained on train nodes (may have noises but are not attacked, e.g. authenticated users), and evaluated on unseen nodes (e.g. new users including fake users). In such a scenario, the *unnoticeability* can be considered by comparing the unseen data with the trusted data.
>
>    Furthermore, we want to refer to the threat model of KDDCUP 2020 "Graph Adversarial Attacks \& Defenses"(https://www.biendata.xyz/competition/kddcup_2020_formal/), the top-tier competition in this field. It considers the injection attack on a large-scale graph, which is similar to our case. The difference is that we revisit the problems in previous works and make some improvements (e.g., changing the transductive setting to an inductive one). All in all, we believe that the chosen threat model has its significance and so far is the most realistic one. Nevertheless, as long as researchers propose other well-defined and realistic scenarios, we will also expand our benchmark to these scenarios.
>
> 2. **Attack constraints**: We agree with you that attack constraints should be included in the main part (it is now in table 5 in appendix), which we are able to do with the additional page. Currently, the attack constraints are: number of injected nodes, number of maximum edges of injected nodes, and range of features of injected nodes. If the injected nodes have too many edges or abnormal features compared to the training nodes, they can be easily detected by defenders. The constraints are set empirically, considering the properties of the original graph (degree distribution, scale of test nodes, feature range of normal nodes). We insist that constraints are not limited to these basic ones. As discussed in Sec 2.4 and 2.5, the attack constraints should not be fixed, attackers should adapt to new constraints if defenders propose a new method. They may need to consider more if they want to defeat incoming defenses. That is also the purpose of the benchmark, which provides a platform for the arms race to go on.
>
> 3. **More challenging task**: Figures 6 and 7 show the results of grb-aminer dataset, which is more difficult to attack due to its large scale. Note that the proposed scenario, due to its black-box and evasion settings, is a challenging task for designing effective attacks. We already include SOTA graph injection attacks like SPEIT (1st-place solution for KDDCUP 2020) and TDGIA (Effective injection attack for large-scale graphs). And we include two strong baselines of defenses to show that the attackers still have a long way to go. Nevertheless, as long as the evaluation setting is unified for both attacks and defenses, the results allow us to compare different methods. We agree with you that the chosen setting might not be as challenging for defenders, we consider introducing other attack settings (e.g. increasing the number of injected nodes) to increase the difficulty of designing robust models.

---

> > ### Comment · Reviewer_fGSw · 2021-07-20
> > **Additional comments**
> >
> > I thank the authors for the reply.
> >
> > I agree that the choice of the threat model allows to standardized evaluation, but, as mentioned above, the main concern is about the lack of works focusing on it, considering all the constraints (black-box, evasion, inductive, injection) at the same time. Then it's not clear how significant it can be for the community, and, about the design choices, if the attack constraints are reasonably challenging.
> > At the same time, the benchmark has a clear structure and allows to reproduce and expand the evaluations.
> >
> > Then, I keep my original score.

---

### Official Review · Reviewer_bbFp · 2021-07-05
**This paper provides a unified, scalable, and extensive benchmark for the robustness of deep neural networks.**

**Rating:** 7
**Confidence:** 4
**Clarity:** Yes, the paper is well written.

**Strengths:**

This paper provides a very rigorous benchmarking of the robustness of graph neural networks. . The benchmark is well documented and leaderboards are easily accessible (both in paper and on the webpage). The paper is also well written and easily accessible.

**Weaknesses:**

This paper does a very good job of benchmarking the robustness of graphs neural networks. Following are my comments to further improve it. (Note that some of them are minor changes, thus do not amount to a weakness. I am listing all of them together to arange them in their order of appearance in the paper )

The majority of research work in adversarial graph learning (consequentially this benchmark too) consider the task of node classification. However, graph neural networks also work well for tasks such as link prediction, clustering, graph classification. It will be helpful if authors can discuss their thoughts on expanding the benchmark to a wider range of tasks.

While the grb framework supports a wider range of configurations, the leaderboard only considers one specific configuration (Table 1). Was there a particular reason authors made this particular choice? Or is it a random selection?

I agree with the authors arguments (section 2.4) about imperceptibility in poisoning attacks in graphs neural networks. However, this section reads more like a discussion, in particular since the benchmark follows a similar setup as previous works. I encourage authors to move this part to discussion.

It's good to have support for external datasets. I also highly encourage authors to integrate open-graph-benchmark/SNAP making it seamless to test attacks/defenses on an even broader range of datasets.

One metric: It's great that authors aggregate the performance of different attacks/defenses with avg/3-min/weighted accuracy. Is it feasible to further reduce it to one number? Having a single number makes it much easier to rank methods (e.g., it unclear which one of three accuracy metrics determines the rank on the leaderboard).

Can authors also provide details on the runtime of different attacks/defenses? This will be helpful to judge whether the grb-aminer dataset is the highest scale for moderate gpu clusters, or whether we can scale up testing robustness of graph nets to an even larger scale.

Disentangle benchmark and algorithmic improvements: I find that the GRB is well designed and well executed. However, authors also propose using layer normalization and adversarial training to improve the robustness of graphs neural networks. Both methods work well, however, I find both their presentation and evaluation pretty thin. For example, the paper is missing a discussion of related work showing how both techniques differ from them, a concrete comparison with previous works, and a more detailed discussion of its results. For example, it's not discussed why adversarial training leads to lower performance in one case (Fig. 9, APPNP). I encourage authors to either 1) move both methods to appendix/discussion 2) Or provide a more detailed discussion on these methods.


**Additional Feedback:**

I've already discussed it in an earlier section.

**Correctness:**

I believe that the evaluation is done correctly and rigorously and the claims are correct.

**Documentation:**

Yes, documentation is available.

**Relation To Prior Work:**

Yes, authors provide adequate references to the related work.

**Summary And Contributions:**

This work provides a unified benchmark of graph neural networks across multiple models, attack vectors, and datasets. It further enables deeper insights by analyzing results through a lens of hardness in attacking different nodes (easy→hard). Further its modular architecture supports multiple different types of attack vectors, external datasets, graph development frameworks.

---

> ### Author Response · Authors · 2021-07-11
> **Thanks very much for your constructive comments.**
>
> Please find our responses to your concerns below:
>
> 1. **Expansion of tasks**: We highly agree that the task of graph learning is not only node classification, but also various tasks you've mentioned. Following most of the previous works in adversarial graph learning, we mainly consider node classification. There do exist few works in the literature about attacks or defenses on these tasks. Since the research on adversarial graph learning is still at a beginning stage (compared to other domains like computer vision), our target is to first establish a benchmark on a single task to attract researchers in the field, then expand it to a wider range of tasks (which of course needs cautions on defining new evaluation scenario, metric, etc.) and include future methods in the literature. In the long term, we will continue to develop and maintain the benchmark.
>
> 2. **Configuration of the threat model and imperceptibility**: The leaderboard currently considers one specific configuration: (black-box, evasion, inductive, injection). This is not a random selection, it is designed for several reasons:
>    * **black-box vs. white-box**: black-box is the most common scenario where attackers can hardly get details of the models. We think it can better reflect the needs for real-world scenarios. Besides, it encourages researchers to consider more general attacks that are effective for various GNNs, instead of attacks that only work when knowing details of specific defenses.
>
>    * **evasion vs. poisoning**: we found that most of previous works follow the setting of poisoning attack in [9], which develops the term of *unnoticeability* or *imperceptibility*. As discussed in Sec 2.4, this setting is controversial. Under the poisoning setting, defenders do not know whether the data are polluted and do not have the original graph. In this case, one can modify the graph largely while not being detected. (quite different from images, which can be easily judged by visual sense. Even there are topological properties for graph, it is difficult to tell whether individual nodes are polluted or not). Thus, there is not yet reasonable constraint or realistic scenario for poisoning attack.
>    * **inductive vs. transductive, injection vs. modification**: to make the *unnoticeability* meaningful and to calibrate the capability of attackers and defenders, we assume that GNNs are trained on training nodes (may have noises but are not attacked, e.g. authenticated users), and evaluated on unseen nodes (e.g. new users including fake users), which naturally makes the scenario an inductive and injection one. In such a scenario, the *unnoticeability* can be considered by comparing the unseen nodes with the training nodes. As discussed in Sec 2.5, we believe this scenario is well-defined and so far the most realistic one. **Nevertheless**, it is definitely not the only scenario, as long as researchers propose other well-defined and realistic scenarios, we will expand our benchmark to them.
>
> 3. **Support for external data**: Thanks for the advice, we are actually working on this to support datasets from OGB, CogDL, or other external resources. Note that the proposed evaluation pipeline is independent of the choice of datasets, it can be easily adapted to any graph data. We processed these five datasets of different scales because they can help better investigate the scalability and diverse properties (different distribution, various domains, etc).
>
> 4. **Choice of metrics**: To be more clear, we provide three metrics that may serve different purposes. For example, if an attacker wants to design a more general attack, he may consider average accuracy (attack works well for most defenses). While if he wants to defeat the most robust defense, he may consider 3-min/weighted accuracy, which attaches more importance to robust defenses (attack sacrifices generality). As mentioned in line 312, the ranking is based on the weighted accuracy since we want to track the most effective methods (for both attacks and defenses). We will emphasize it both in the paper and on the website.
>
> 5. **Runtime of methods**: We agree that it is meaningful to provide the runtime of methods, and we are working on it. All current experiments can be operated on a single V100 GPU with 32GB RAM to give you an idea.
>
> 6. **The proposed layer normalization and adversarial training**: We agree that these parts currently seem thin. Due to the specificity of this benchmark track, we use most of the content on introducing how the benchmark is designed and the related framework. The reasons that we include these two methods are: (1) They are all standard methods that can be easily adapted to the proposed scenario while being effective. (2) Previous attacks are not as effective as they claimed when some defense tricks are added to GNNs. (3) They can serve as strong baselines to promote future research. We will elaborate this part in the additional page allowed for the final version.

---

> > ### Comment · Reviewer_bbFp · 2021-07-20
> > **Additional comments**
> >
> > I thank authors for providing a detailed response to my comments. It clarifies the concerns I had in my review and also incorporate the suggestions (such as reporting runtime, more datasets, etc.). I find that the paper will be a valuable addition to the community, thus I am keeping my original recommendation intact.

---

### Decision · Program_Chairs · 2021-07-26

**Decision:**

Reject

**Comment:**

The paper received mixed reviews, with two reviewers still leaning towards rejecting the paper after taking the authors' response into account. There were two main concerns:

(A) Is the submission in scope for the benchmarks track? In particular, Reviewer 16ob views the role of the benchmark track primarily as a venue for work on datasets and how to conduct benchmarks (frameworks, etc.), rather than on specific benchmarks.

(B) Is the specific setting / threat model in the submission actively studied? Reviewer fGSw points out that adversarial robustness has been studied before in the context of graph learning, but no prior work conforms to the specific threat model in the submission.

Overall this is a difficult decision since for instance Reviewer 16ob agrees that for the specific threat model studied, the paper represents a high quality contribution.

My perspective is as follows: I consider the submission to be in scope for the datasets & benchmarks track since proposing and carefully executing a benchmark for a specific problem is important foundational work in machine learning. However, the specific problem investigated via the benchmark should be relevant and its relationship to other benchmarks should be clear. In the current paper, this is an area for improvement since the specific threat model has not been studied in prior work yet (Reviewer's fGSw point). In addition, the submission does not contain experiments relating the proposed threat model to other threat models already studied in the literature. It would be particularly important to know how the proposed threat model relates to the KDD Cup benchmark the authors mention. For instance, does the ordering of attacks or defenses change between the two benchmarks? Do the high-level conclusions from the two benchmarks agree?

So unfortunately my recommendation is to reject the paper from the first round of the datasets & benchmark track. Overall I find the paper already an interesting contribution, and after taking the reviewers' comments into account, the paper will be a strong submission for the second round of the datasets & benchmarks track or another venue.